# Mental health challenges among adolescents and young adults with perinatally acquired HIV: Key findings from the I'mPossible program in India

Ashley A. Sharma[1], Michael Babu Raj[2], Babu Seenappa[2], Siddha Sannigrahi[3], Kacie Filian[3], Esha Nobbay[4], Suhas Reddy[2], Prashant Laxmikanth[2], Sanya Thomas[5], Aastha Kant[6], Satish Kumar S. K.[7], Sunil S. Solomon[8], Lakshmi Ganapathi[5‡], Anita Shet[3‡*]

**1** Robert Larner, M.D. College of Medicine, University of Vermont, Burlington, Vermont, United States of America, **2** RISHI Foundation, Bengaluru, India, **3** Department of International Health, Johns Hopkins Bloomberg School of Public Health, Baltimore, Maryland, United States of America, **4** Department of Psychiatry, Adichunchanagiri Institute of Medical Sciences, B G Nagara, India, **5** Department of Pediatrics, Boston Children's Hospital and Harvard Medical School, Boston, Massachusetts, United States of America, **6** Johns Hopkins India Private Limited, New Delhi, India, **7** Y.R. Gaitonde Center for AIDS Research and Education, Chennai, India, **8** Division of Infectious Diseases, Johns Hopkins School of Medicine, Baltimore, Maryland, United States of America

☉ These authors contributed equally to this work.
‡ These authors contributed equally to this work.
* ashet1@jhu.edu

## Abstract

Adolescents and young adults with perinatally acquired HIV (APHIV) face elevated risk for common mental health disorders (CMD). To understand determinants of CMD among APHIV in India, we conducted a mixed methods study by screening for depression and anxiety among a cohort of APHIV in southern India. Between March-June 2023, we administered a cross-sectional survey to participants in the I'mPossible Fellowship, a peer-led mentorship program. We incorporated participatory research principles, wherein trained peer mentors (youth investigators) iteratively designed and administered focus group discussions and screening tools for depression (Patient Health Questionnaire-9: PHQ-9), anxiety (Generalized Anxiety Disorder-7: GAD-7), resilience (Child Youth Resilience Measure-Revised – CYRM-R), and an abbreviated HIV stigma Scale. We used multivariable regression to identify correlates of positive CMD screens and inductively analyzed focus group transcripts. Among 185 APHIV survey participants, mean age was 18.6 years (SD 3.5 years); 63.2% were male. Most (91.9%) had lost one or both parents, and 43.2% lived in child-care institutions (CCIs). Majority (90.4%) were virally suppressed (VL < 150 copies/mL). A high proportion screened positive (severity category of mild and above) for at least one CMD (62.7%), depression alone (25.9%), GAD alone (7%), or both (29.7%). Corresponding prevalence for moderate or severe categories were 18.9%, 14.6%, 9.7%, and 5.4%, respectively. Externalized stigma was high (74.6%),

**Data availability statement:** The de-identified dataset supporting the findings of this study is provided as supporting information and is publicly available in accordance with PLOS data sharing policies.

**Funding:** This work was supported by National Institutes of Health (K23DA057151 to LG and AAS), Rishi Children's Fund (MBR, BS, EN, SR, PL), Department of International Health, Johns Hopkins Bloomberg School of Public Health (SS, AS), Harvard University Center for AIDS Research (ST).

**Competing interests:** The authors have declared that no competing interests exist.

reinforcing disclosure concerns (81.1%). Loss of both parents was associated with increased odds of anxiety (aOR 2.10, 95% CI 1.07- 4.09). Exploration of anxiety and depression-related factors revealed themes that included uncertainty about transitioning to adult care, ART adherence challenges, and maladaptive coping mechanisms. Family support, disclosure fears, school pressures, stigma, and evolving societal attitudes shaped participants' mental health experiences. Our findings underscore the need to prioritize integration of mental health screening and interventions across the pediatric-to-adult HIV care continuum in India.

## Introduction

As of 2022, 1.65 million adolescents aged 10–19 years and 3.1 million young people aged 15–24 years were living with HIV worldwide [1,2]. Collectively, adolescents and young adults (defined as those between the ages 10–24 years, referred to as 'youth' in this paper) constitute at least a third of all new HIV infections globally [2–4]. Adolescence is a distinct developmental phase characterized by significant physical, psychological, and social changes, marking a critical transition from childhood to adulthood [5,6]. This period is associated with both opportunities for social and emotional growth, as well as a myriad of social, emotional, and behavioral challenges [7]. Approximately 50% of mental health disorders are estimated to have onset during adolescence [8,9], and they contribute to suicide, a leading cause of death among adolescents in all regions of the world [10–12]. Multiple studies indicate that young people with HIV, particularly adolescents and young adults with perinatally acquired HIV (APHIV) have a high prevalence of common mental health disorders (CMD) such as depression, generalized anxiety disorder (GAD), and post-traumatic stress disorder (PTSD) [13–16]. Systematic reviews estimate that 26.1% of youth with HIV have depression [15], while 45.6% have anxiety [17]. The burden of CMD is higher among this population compared to youth without HIV and is influenced by distinct physiological and psychosocial vulnerabilities, including the physiological effects of HIV and treatment regimens, the burden of managing a chronic illness [18], and psychosocial stressors such as pervasive stigma and discrimination [19–22].

CMD have a significant impact on HIV treatment outcomes; co-existing unaddressed depression and/or anxiety disorders are associated with decreased antiretroviral therapy (ART) adherence, increased sexual risk behaviors, lower viral suppression, higher attrition across the HIV treatment cascade, and increased mortality rates [23–26]. While ART has contributed to decreased HIV-related mortality overall [27–29], this double burden of the HIV epidemic and CMD is poorly recognized and addressed in low- and middle-income countries (LMICs), where approximately 90% of APHIV live [30,31]. Studies evaluating CMD among APHIV in LMICs have primarily focused on countries in sub-Saharan Africa and to a lesser extent, in Southeast Asia. Such research is limited in India [32,33] although a substantially large APHIV population live in India, with numbers comparable to countries in sub-Saharan Africa [34–36].

Globally there is a paucity of youth-tailored evidence-based mental health interventions for CMD among APHIV [37]. More recently interventions spanning family-based psychosocial and economic interventions, as well as group and community-based interventions have been developed in pilot studies and randomized controlled trials among youth with HIV [37–39]. Some of these interventions include elements of cognitive-behavioral therapy (CBT), mindfulness, problem-solving therapy, and other evidence-based mental health interventions [39–43]. In India's context, the lack of well-studied mental health interventions specifically tailored for APHIV is stark. There is a great need for interventions for preventing and addressing CMD to be integrated into public sector HIV treatment programs, where most APHIV seek care.

In 2021, we established the I'mPossible Fellowship, an 18-month program in which young adults living with HIV, termed 'fellows' receive training and supervision to provide support related to ART adherence, education, and psychosocial needs to children and adolescents with HIV, termed 'peers', through one-on-one and group mentorship. While peer support can enhance HIV and broader psychosocial outcomes [40–43], addressing specific CMD necessitates data-informed contextual evidence-based interventions. In this study, we aimed to screen for CMD among APHIV participants of the I'mPossible Fellowship in India, explore their psychosocial experiences, and understand structural and social determinants of CMD to inform the development of youth-tailored mental health interventions.

## Methods

### Ethics statement

This study obtained approval from the Institutional Review Boards of the Y.R. Gaitonde Centre for AIDS Research and Education (YRGCARE), Chennai, India and the Johns Hopkins Bloomberg School of Public Health, Baltimore, Maryland, USA.

### Participants and study setting

The I'mPossible Fellowship program was established in the southern states of Karnataka and Tamil Nadu through collaborative efforts with child care institutions (CCIs) and community-based organizations providing integral support, such as safe residential care, medical treatment, education, and psychosocial assistance, to children and APHIV in institutional and family-based care, ages 8–26 years (peers). Trained youth living with HIV, ages 18–27 years (fellows), provide this support for peers through one-on-one sessions and group sessions over a 48-week period, in the critical domains of health, education, livelihood, and gender. Fellows often reside in partner CCIs or maintain regular contact with peers in family-based care through phone calls. Their role also includes monitoring ART adherence and addressing a range of challenges peers may face, including academic difficulties, emotional distress, or barriers in accessing healthcare services [44].

Eligible participants in this study were peers of the program, between ages 15–24 years, received a diagnosis of HIV prior to 10 years of age, were resident in CCIs or in family-based care. Interventions integral to the I'mPossible Fellowship program are described in detail elsewhere [45]. In addition to the regular structured interventions within the program, CCI supervisors were available to provide oversight and facilitate referrals to state welfare schemes, pediatric specialists, psychiatrists, and other mental health professionals. Although it provides well-rounded support to APHIV, the I'mPossible program in its current form does not incorporate evidence-based interventions to address CMD among peers, and higher level needs such as treatment of CMD are primarily addressed through referrals. Exploring CMD and psychosocial experiences in this study was a crucial step towards developing targeted interventions addressing mental health in this population.

We conducted the study between March and June 2023. Participants included peers from the first cohort of the I'mPossible Fellowship that comprised 257 peers. Those ≥13 years were included in this study. All had a diagnosis of HIV prior to age 10 years (definition of perinatally-acquired HIV) and were aware of their HIV status. For participants ≥18 years, informed consent was obtained by trained research staff, and for participants <18 years, consent from a parent or legal guardian followed by written assent from the minor participant was obtained.

## Youth investigators and community-based participatory research

For survey development, adaptation, and administration, we incorporated a community-based participatory approach by including 5 fellows from the second cohort and 4 fellows from third cohort of the I'mPossible Fellowship (who had not directly provided mentorship to study participants) [44]. These 'youth investigators' actively shaped the research process by culturally adapting the English, Tamil, and Kannada versions of CMD screening instruments utilized in the survey through iterative discussions with experts, while simultaneously drawing on their own perspectives as APHIV. Prior to conducting study assessments, they also underwent certification in human subjects' protection, training in survey administration techniques, and mental health screening, including recognizing when peer participants needed referrals to counselors or mental health professionals for further evaluation and care.

## Mental health surveys

The comprehensive mental health survey comprised a general questionnaire and specific instruments to measure CMD, resilience, and stigma; each of these components were previously validated in several low-and-middle income countries, including India [46–49]. The survey components are described below:

(i)　The general questionnaire collected the following data: sociodemographic characteristics, such as age, gender, place of residence; education characteristics, including current school enrollment status and reasons for school discontinuation; employment characteristics, including current employment status and workplace challenges; and HIV treatment characteristics, such as self-reported ART adherence, and viral load measurements taken within the past year.

(ii) The Child and Youth Resilience Measurement (CYRM-R). [50] This screening tool was designed to identify strengths and resources possessed by participants, assessing the quality of peer participants' relationships with family and caregivers, and inner strengths, such as emotional regulation and problem-solving skills [50,51].

(iii) The Patient Health Questionnaire (PHQ-9) Questionnaire [49,52]. To screen for depression among peer participants, we administered the PHQ-9, a 9-item questionnaire using a Likert scale scoring system to categorize depressive symptoms over the past two weeks into five severity levels: none/minimal (score 0–4), mild (score 5–9), moderate (score 10–14), moderately severe (score 15–19), and severe (score 20–27).

(iv) The General Anxiety Disorder-7 (GAD-7) Questionnaire: [49,53] We also administered the 7-item GAD-7 questionnaire categorizing symptoms of GAD over the past two weeks into four severity levels: minimal (score 0–4), mild (score 5–9), moderate (score 10–14), and severe (score 15–21).

(v) HIV Stigma Questionnaire: To assess peer participants' perceptions of HIV-related stigma, we created an abbreviated 4-item version of a 12-item HIV Stigma Survey [54], which was previously validated among adolescents with HIV in India [55], and itself adapted from Berger's validated 40-item stigma questionnaire [56,57]. The abbreviated version was intended to obtain initial insights of HIV-related stigma and comprised four statements, representing constructs of internal stigma (personalized stigma, and negative self-image), and external stigma (concerns about disclosure, and concerns regarding public attitudes towards individuals with HIV) (Table 1) [55,57,58].

## Focus group discussions

We conducted three FGDs, each consisting of six participants (n = 18), recruited across three sites, Belgaum, Bangalore, and Krishnagiri. Aiming to explore peers' psychosocial experiences, we used purposive recruitment strategies to include equal representation from adolescents <18 years of age and young adults between ages 18–24 years of both genders, and

**Table 1. Sociodemographic characteristics, HIV treatment, resilience, and stigma measures of adolescents and young adults with perinatally acquired HIV (n = 185).**

| Sociodemographic Characteristics | | n (%) |
|---|---|---|
| Age, years | Mean (SD) | 18.6 (3.5) |
| Gender | Male | 117 (63.2) |
| Place of residence | Rural | 101 (54.6) |
| Current resident in child care institution | Yes | 80 (43.2) |
| Primary caregiver in household (n = 103) | One parent | 46 (44.7) |
| | Both parents | 11 (10.7) |
| | Grandparents | 16 (15.5) |
| | Non-parents first degree relative | 11 (10.7) |
| | 2nd and 3rd degree relatives | 16 (15.5) |
| | Non-relative | 3 (2.9) |
| Parents' status | Both deceased | 93 (50.3) |
| | Only one parent alive | 77 (41.6) |
| | Both alive | 15 (8.1) |
| Status of current education | Currently in school | 130 (70.3) |
| | Discontinued education after 10th grade | 35 (18.9) |
| | Discontinued education after 12th grade | 20 (10.8) |
| Reasons for discontinuation among those who are not currently studying (n = 65) | Unable to get a passing grade | 15 (23.1) |
| | Parents passed away | 6 (9.2) |
| | Lacking financial support | 29 (44.6) |
| | Needed to find work for supplemental income | 15 (23.1) |
| Currently employed | Yes | 74 (40.0) |
| | No | 111 (60.0) |
| Challenges faced in the workplace | Not a good fit | 4 (5.4) |
| | Unable to take care of health needs | 7 (9.6) |
| | Insufficient income to meet needs | 21 (28.8) |
| | External disclosure concerns | 4 (5.5) |
| | Other (challenges with transportation and time management) | 10 (13.7) |
| **HIV Treatment** | | |
| Duration on ART (years) (n = 179) | Median (IQR) | 9.2 (6.1-13.1) |
| Viral load reported in past 12 months of survey (copies/mL) (n = 178) | Not detected (viral load<150) | 161 (90.4) |
| Self-reported adherence category (n = 184) | Optimal Adherence (0-<7 days missed) | 151 (83.0) |
| **Resilience** | | **Median (IQR)** |
| Total resilience score (max score 85) | | 74 (69-78) |
| Low resilience score (≤25th percentile) (n) | | 51 (27.6) |
| **Stigma** | | **n (%)** |
| Concerns about public attitudes | | 138 (74.6%) |
| Disclosure concerns | | 150 (81.1%) |
| Personalized stigma | | 89 (48.4%) |
| Negative self-image | | 35 (18.9%) |

from APHIV living in child care institutions and those living in family-based care. The discussion guide for the focus group discussions was developed by the research team with significant input from current and former fellows and informed by the socioecological model which recognizes the multifaceted and dynamic inter-relatedness between individual and environmental factors including family, school, community and policies/governmental agencies that impact health and behavior [59]. Within this framework, questions were formulated to assess factors contributing to overall mental health, such as peer participants' personal and interpersonal psychosocial experiences, including experiences within and perceptions regarding their community and the larger society, existing supports, and coping behaviors. FGDs were facilitated by a trained fellow from the third cohort, with the participation of a study investigator (with MD or PhD training) who had experience facilitating focus groups. They were conducted in a private space and had an average duration of 1 hour. Focus groups for female adolescents and young adults were conducted separately from focus groups for male adolescents and young adults. Facilitators were gender concordant with focus groups. The discussions' audios were recorded and translated from Kannada/Tamil to English to allow for qualitative coding of the translated transcripts. Additionally, we conducted member checking of transcripts to ensure the accuracy of the findings by reviewing them with the peer participants.

## Data definitions

**Resilience.** For the resilience measure, we divided scores into quartiles such that a total CYRM-R score of ≤25th percentile was defined as 'low resilience'. As resilience tends to vary widely in different contexts, rendering a fixed threshold would be impractical. As per reccommendations from the tool developers, we divided the scores into quartiles and picked the lowest quartile (scores below the 25th percentile of all scores) to be defined as "low resilience" scores. [60] This method is supported in the broader literature as a reasonable way to operationalize lower resilience in non-clinical, population-based samples. For sensitivity analysis, we assessed ≤15th, ≤33rd and ≤50th percentile of the total CYRM-R score, to determine if a change in threshold materially impacted the results.

**Screening for depression and anxiety.** The presence of a positive screen for depression was defined as a severity of mild and above (score of ≥5) according to the PHQ-9, [52] and presence of a positive screen for GAD was defined as a severity of mild and above (score of ≥5) based on the GAD-7 [61]. These cut-offs were chosen for several reasons: (i) Prior community-based studies in India among adolescents in India have utilized similar cut-odds for initial screening of CMD [62–64], (ii) Existing evidence suggests that individuals with mild depression and/or anxiety are at risk of progression to moderate or severe illness with estimates ranging from 2% to 45% [65–67]. Therefore, among adolescents, guidelines suggest close monitoring as well as augmented supports, including peer support for individuals who have positive screens for mild depression and/or anxiety [68]. This is particularly relevant for our population of peers within the context of the I'mPossible Fellowship as identification of peers with positive screens for mild depression and/or anxiety in addition to those with positive screens for more severe depression and/or anxiety enables us to identify additional ways to augment existing support for these peers.

**HIV parameters.** Detectable viral load was defined as ≥150 copies/mL, consistent with established literature [69,70]. Self-reported ART adherence in the past month was categorized as optimal (0-<7 days of medications missed) or suboptimal (≥7 days of medications missed).

## Data analysis

**Quantitative.** We characterized the study cohort using summary statistics and generated multivariable logistic regression models to determine correlates of a positive screen for (1) depression alone, (2) GAD alone, and (3) at least one CMD (i.e., depression and/or anxiety). Variables included (a) sociodemographic characteristics (age, gender, place of residence); (b) education and employment characteristics (school discontinuation and employment status); (c) HIV treatment characteristics (duration on ART and viral suppression); (d) psychosocial factors (low resilience and parental deceased status). Variables assessed in univariable analysis were included in the multivariable model based on prior evidence of likely association with CMD. [8,71–81] Analyses were performed using SPSS version 29.0 (IBM Corp, Armonk, New York).

**Qualitative.** Analysis of the FGDs' transcripts entailed inductive and deductive approaches for thematic analysis. Initially, a coding scheme was developed through collective team discussion and insights from comparable studies. Subsequently, focus group discussions transcripts were independently coded using Dedoose software by two researchers (AAS and AK), who then convened to identify emerging themes, address discrepancies, and finalize the coding scheme. A third researcher (EN) evaluated the transcripts, overseeing the reconciliation process and ensuring consistent application of codes.

## Results

### Survey findings

**General characteristics of participants.** Among 216 age-eligible peers who were contacted for the mental health survey administration, 185 peer participants responded (12 could not be contacted, 19 cited lack of time or interest for declining). Survey participants had a mean age of 18.6 years (SD 3.5 years) and 63.2% were male, 50.3% reported that both parents were deceased, 54.6% lived in rural areas, and 43.2% lived in child care institutions (CCIs). Among those who lived in family-based care, 44.7% lived in one-parent households. Educational discontinuation was reported by 29.7%, the primary reasons being financial constraints (44.6%) and challenges in keeping up with schoolwork (23.1%). With regard to employment, 40.0% of the participants were currently working but reported facing challenges, including insufficient income (28.8%) and fears of HIV disclosure in the workplace (5.5%) (Table 1).

**HIV treatment characteristics.** Participants' mean duration on ART was 9.7 years (SD 4.0 years) (Table 1). Viral suppression (viral load <150 copies/mL in the past 12 months since study recruitment) was high at 90.4%. Optimal adherence was seen among 83.0% of participants who reported <7 days of missed doses in the past month.

**Resilience measures.** Total resilience score and subscale scores were high among the peer participants (Table 1). Low resilience (CYRM-R score of ≤25th percentile) was observed in 51 (27.6%) participants.

**Stigma survey.** The majority of peer participants, 74.6%, reported that people with HIV face rejection when their status is disclosed, reflecting concerns about public attitudes toward individuals with HIV. Similarly, 81.1% reported making considerable efforts to keep their HIV status, or that of their parents, a secret, highlighting deep concerns about external disclosure. Nearly half, 48.4%, reported being shunned by people who became aware of their HIV status or that of their parents. However, perceptions of negative self-image were comparatively low, with 18.9% of peer participants reporting such feelings.

**Screening for depression and generalized anxiety among APHIV.** Among all APHIV participants, 62.7% screened positive for "any" (at least mild and above) depression and/or generalized anxiety, with 18.9% meeting criteria for "significant" (moderate or more severe) symptoms. While 55.7% screened positive for any level of depression and 14.6% for significant depression; 36.8% screened positive for any level of GAD and 9.8% for significant GAD (Table 2).

Double orphan status emerged as a significant correlate (aOR: 2.10, 95% CI: 1.07-4.09, p = 0.030) in multivariable analysis of a positive screen for any anxiety (GAD-7 score of ≥ 5), (Table 3). In univariable and multivariable analyses of a positive screen for any depression alone or any CMD, no correlates showed statistically significant associations. Analyses examining moderate and above categories of CMD (score ≥10 for depression and/or GAD) yielded no additional significant associations. Sensitivity analyses of low resilience (using thresholds of ≤15th, ≤ 33rd, and ≤50th percentiles) in separate models showed similar findings.

### Insights from focus group discussions

Among the 18 FGD participants mean age was 16.7 years (SD = 1.5), 12 were male adolescents or young adults, 15 resided in childcare institutions, and all were studying or employed. Among the 12 FGD participants who had also completed the mental health survey, 6, 4 and 7 screened positive for depression, GAD and at least one CMD, respectively, highlighting mental health vulnerabilities within this group. Data saturation was reached for all major topics. Several themes emerged across the

**Table 2. Prevalence of positive screens for depression and general anxiety disorder among adolescents and young adults with perinatally acquired HIV (n = 185).**

| Common mental health disorder (CMD) | Category [scores] | n (%) |
|---|---|---|
| Depression | None/minimal [0–4] | 82 (44.3) |
| | Mild [5 –9 ] | 76 (41.1) |
| | Moderate [10 –14] | 25 (13.5) |
| | Moderately severe [15 –19 ] | 2 (1.1) |
| | Severe [20 –27 ] | 0 (0.0) |
| Any depression | Categories mild and above [≥5] | 103 (55.7) |
| Significant depression | Categories moderate and above [≥10] | 27 (14.6) |
| Generalized Anxiety Disorder (GAD) | Minimal [0–4] | 117 (63.2) |
| | Mild [5 –9 ] | 50 (27.0) |
| | Moderate [10 –14] | 14 (7.6) |
| | Severe [>=15] | 4 (2.2) |
| Any anxiety | Categories mild and above [≥5] | 68 (36.8) |
| Significant anxiety | Categories moderate and above [≥10] | 18 (9.8%) |
| Screening for Common Mental Health Disorders | Both Depression and GAD | 55 (29.7) |
| | Depression, no GAD | 48 (25.9) |
| | GAD, no Depression | 13 (7.0) |
| | Depression and/or GAD | 116 (62.7) |

socioecological model (SEM) constructs that FGD participants identified as potential contributors or alleviators of symptoms of anxiety and/or depression. Narratives of stigma crossed multiple domains of the SEM. (Fig 1 and Table 4).

**Individual factors**

**Uncertainty with transitioning into adult care.** Participants' concerns spanned across education, employment, ART adherence challenges, and future support after transitioning out of known care systems. (Table 4). APHIV who lived in CCIs expressed worries about navigating the health system after they left these institutions and in general described being underprepared for transition.

**HIV treatment adherence challenges.** While participants generally had high ART adherence, they identified situations where taking daily medications was challenging (for example, when at work, or when traveling), and found taking medications to be a stressful experience due to persistent concerns around inadvertent disclosure.

**Individual coping mechanisms.** Participants relied in part on individual coping mechanisms, such as distraction through hobbies and self-care activities. However, particularly among young adults who had transitioned from CCIs to independent living, participants described maladaptive coping mechanisms with substance use in order to deal with structural and psychosocial vulnerabilities including financial and housing insecurity, and poor social support.

**Interpersonal factors**

**Relationships with friends and families.** Family support played a significant role as a protective interpersonal factor, whereas lack of family support was a contributor to symptoms of anxiety and/or depression. There were notable differences in family support between APHIV living in CCIs and those living in the community; the former were often double orphans and primarily relied on a sibling or other peers in the CCIs for support. They perceived their extended family members as lacking understanding, whereas APHIV living in the community received support from a parent (usually of shared HIV status) that they could easily confide in.

**Table 3. Factors associated with positive screens for common mental health disorders[a] among adolescents and young adults with perinatally acquired HIV (n = 185).**

| | Generalized Anxiety Disorder | | Depression | | At least one common mental health disorder | |
|---|---|---|---|---|---|---|
| | Univariate model OR (95% CI) | Multivariate model aOR (95% CI) | Univariate model OR (95% CI) | Multivariate model aOR (95% CI) | Univariate model OR (95% CI) | Multivariate model aOR (95% CI) |
| **Age** (continuous) | 1.02 (0.93-1.11) | 0.96 (0.85-1.08) | 1.02 (0.94-1.11) | 1.00 (0.89-1.12) | 1.03 (0.94-1.12) | 1.00 (0.89-1.12) |
| **Gender** | | | | | | |
| Woman | 1.64 (0.89-3.04) | 1.44 (0.72-2.91) | 1.35 (0.74-2.47) | 1.17 (0.59-2.29) | 1.56 (0.83-2.93) | 1.37 (0.68-2.75) |
| Man (Ref) | | | | | | |
| **Parents' status** | | | | | | |
| Both parents deceased | 1.73 (0.94-3.16) | 2.10 (1.07-4.09)* | 1.22 (0.68-2.17) | 1.17 (0.62-2.19) | 1.70 (0.93-3.10) | 1.69 (0.89-3.24) |
| One or both parents alive (Ref) | | | | | | |
| **Place or residence** | | | | | | |
| Rural | 1.09 (0.60-1.98) | 0.99 (0.50-1.95) | 0.82 (0.46-1.47) | 0.77 (0.40-1.49) | 0.97 (0.53-1.77) | 0.87 (0.44-1.71) |
| Urban (Ref) | | | | | | |
| **Living in a child care institution** | | | | | | |
| Yes | 0.88 (0.48-1.60) | 0.89 (0.41-1.97) | 0.80 (0.44-1.43) | 1.03 (0.49-2.19) | 0.99 (0.54-1.80) | 1.26 (0.58-2.74) |
| No (Ref) | | | | | | |
| **Discontinuation of school** | | | | | | |
| Yes | 1.51 (0.79-2.88) | 1.47 (0.68-3.18) | 1.29 (0.68-2.44) | 0.97 (0.46-2.04) | 1.33 (0.68-2.58) | 1.12 (0.52-2.40) |
| No (Ref) | | | | | | |
| **Currently working[‡]** | | | | | | |
| Yes | 1.44 (0.70-2.96) | 1.50 (0.59-3.86) | 1.03 (0.51-2.08) | 0.74 (0.31-1.78) | 1.17 (0.57-2.40) | 0.85 (0.34-2.10) |
| No (Ref) | | | | | | |
| **Complete Viral Suppression** (VL < 150) | | | | | | |
| No | 0.92 (0.32-2.61) | 1.21 (0.39-3.75) | 1.52 (0.54-4.31) | 1.14 (0.38-3.45) | 1.12 (0.39-3.18) | 0.85 (0.28-2.61) |
| Yes (Ref) | | | | | | |
| **Duration on ART** (months) | 1.00 (0.99-1.01) | 1.00 (0.99-1.01) | 1.00 (1.00-1.01) | 1.00 (0.99-1.01) | 1.00 (1.00-1.01) | 1.00 (1.00-1.01) |
| **Total resilience** | 1.16 (0.60-2.46) | 0.79 (0.37-1.69) | 1.88 (0.96-3.70) | 1.68 (0.81-3.52) | 1.84 (0.91-3.72) | 1.37 (0.64-2.95) |
| ≤ 25th centile | | | | | | |
| > 25th centile (Ref) | | | | | | |

[a] Common mental health disorders defined as positive screens for depression and/or anxiety of severity category of mild and above.

[‡] Model only includes those ≥ 18 years of age

**Disclosure concerns.** Personalized stigma (a community factor) intersected with disclosure concerns (an interpersonal factor). Several participants expressed fears around others avoiding them or not wanting to talk to them or interact with them because of their HIV status. As a result, participants generally avoided disclosing their status to even people who knew them well.

**Interpersonal resilience.** Participants identified family and friends as key sources of emotional support, providing a comfortable space for sharing their feelings. Among family members, siblings, both with and without HIV, were often mentioned as individuals with whom participants spoke to and supported them the most. Some participants noted

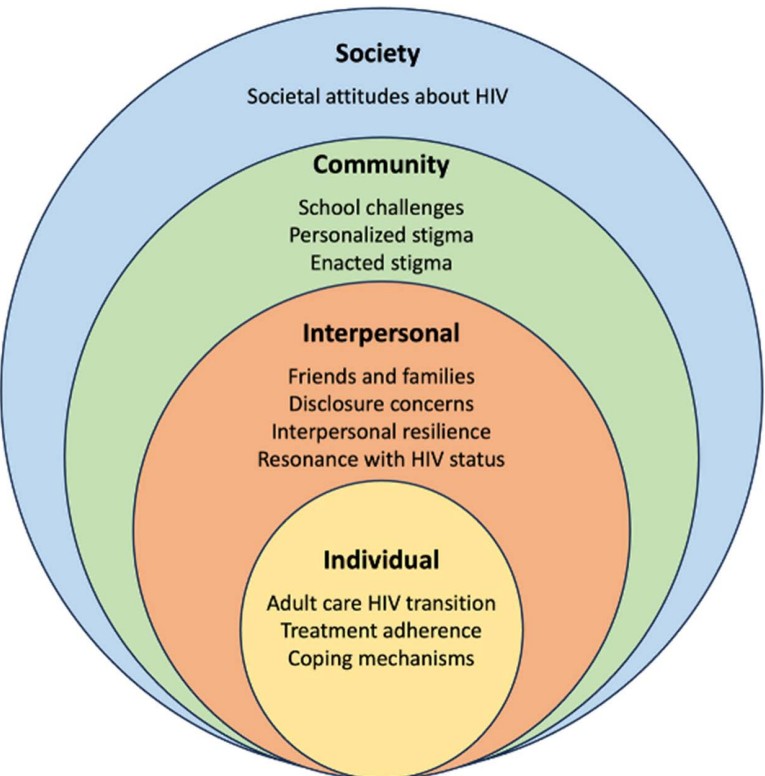

**Fig 1. Categorization of themes based on the Socio Ecological Model (SEM) domains.**

that family members who did not have HIV often struggled to understand their stress. Conversely, some participants expressed distrust towards their friends, fearing loss of confidentiality.

**Resonance with HIV status.** When discussing fellows who shared their HIV status, participants expressed that they felt no hesitation in sharing their challenges given mutual understanding. They often mentioned that it was easier to talk to someone who understood their situation compared to someone who did not. In particular, when it came to issues related to HIV health, such as hospital visits and obtaining ART, fellows proved to be vital sources of support.

### Community factors

**School challenges.** School was identified as a significant source of distress, stemming from various factors, including academic pressure, fear of others discovering their status, and lack of support to continue their education. Schools were also primary spaces where participants interacted with others without HIV, where they often faced challenges when their status was known, including having to move schools. Sources of anxiety included daily studies and exams, and financial support needed to continue their studies.

**Personalized and enacted stigma.** Participants expressed a strong belief that they would face rejection, isolation, or negative treatment from others due to their HIV status, leading to increased anxiety. This concern was particularly evident when discussing social interactions and general societal attitudes where participants feared that disclosure of their status could lead to social distancing or exclusion from events. These anxieties impacted their participation in activities and school attendance.

**Table 4. APHIV participants' quotations from focus group discussions.**

| Themes | Quotations |
|---|---|
| **Individual** | |
| Uncertainty with transitioning into adult care | "After finishing 12th standard, we are not sure if we will stay in the care institution or not and if, at that time, they suddenly send us out, we cannot find any job and we will be alone. We are not aware what support we will be getting. What will be our future?" – *CCI resident* |
| HIV treatment adherence challenges | "Some who are working on night duty find it scary to take tablets when people are around." – *CCI resident* |
| Individual coping mechanisms | "[People] drink because of family issues. And stress about their future. They are not able to find jobs" – *CCI resident* |
| **Interpersonal** | |
| Relationships with friends and families | "Among our [childcare institution] friends [stress is] not there but with families they are not able to understand" – *CCI resident* |
| Disclosure concerns | "If they come to know that we have HIV then, they will stop talking to us and keep us far from them." – *CCI resident* |
| | "If I talk about my status, they might not talk to me again, and I am scared" – *Resident in family-based care* |
| Interpersonal resilience | "I share my problems with friends and family members." - *Resident in family-based care* |
| Resonance with HIV status | "With HIV positive people, we can speak much more freely. I can't speak to unknown people." – *Resident in family-based care* |
| **Community** | |
| School challenges | "I was struggling with studies, and then later I planned to discontinue from schooling." – *Resident in family-based care* |
| | "The school does not know about my status." - *Resident in family-based care* |
| Personalized stigma | "I am bit scared of doing education in regular schools because if all the students in the school are negative how will people treat me." - *CCI resident* |
| | "I think they [people in society] might avoid me if they know I have HIV and will not talk and be close to me. That is a fear I have." – *Resident in family-based care* |
| Enacted stigma | "Through my mother almost all my village people got to know that we are infected…Still there are people who avoid me." - *CCI resident* |
| | "We were in another school and when they found out [about our HIV status], they threw us out." – *CCI resident* |
| **Society** | |
| Societal norms influencing attitudes towards people with HIV | "Some people look at us differently, but others do not." - *Resident in family-based care* |
| | "When I was a kid, many people were avoiding me and not talking to me, but now the situations are changing, people are getting to know the mode of HIV infection spread is not like COVID. Now the people of my village talk to me well."- *CCI resident* |

## Societal factors

**Societal norms influencing attitudes towards people with HIV.** Participants had more nuanced perceptions about societal and public attitudes towards people with HIV. While they observed a broader climate of stigma and negative views, they recognized that not all individuals embraced these attitudes. For example, one participant noted how, despite some community members avoiding them due to HIV status, others began to engage more positively as awareness about HIV transmission increased leading to greater acceptance. These experiences reflect that societal attitudes do not permeate monolithically into personal interactions, leading to more individualized perceptions of acceptance or rejection.

## Discussion

Within a contemporary context of widespread ART availability and accessibility in India, our study shows that nearly two-thirds of APHIV screened positive for depression and/or anxiety, and over three-quarters reported externalized stigma and

disclosure concerns, highlighting their unmet mental health vulnerabilities despite achievement of treatment success with high rates of viral suppression. Our estimates of prevalence of positive screens for CMD among APHIV have implications for youth in India regardless of achievement of HIV treatment success. Despite more than 90% of APHIV in our study being virally suppressed, the high burden of mild or greater severity of CMD, suggesting the need for interventions that extend beyond viral suppression alone. The prevalence of positive screens for moderate and severe depression (14.6%) and moderate and severe anxiety (9.8%) in our cohort is comparable to studies among similar populations in other settings with high levels of viral suppression. [82–84] For example, in a recent study in Thailand among a cohort of 100 young people with HIV, of whom 81.0% had undetectable viral loads, approximately 20% experienced positive screens for significant depression or anxiety [84]. In particular, 7.0% had isolated depression, defined by a modified PHQ-9 for adolescents score of ≥10 (moderate severity or higher); 2% had isolated anxiety, defined by a Child Anxiety Related Disorder (SCARED) score of ≥25, thus indicating the presence of an anxiety disorder; 8% experienced comorbid anxiety and depression. [84] External comparisions of the estimates of positive screen for depression and anxiety in our analyses with studies among adolescents in India is challenging given the high variability in both screening measures and prevalence of depression and anxiety across different regions and adolescent populations (school going versus non-school going, rural versus urban, younger versus older adolescents) [85,86]. However, specific to depression, estimates in our study compared to findings among school-going adolescents in a similar age band living in a district with geographic overlap were higher for both mild depression alone (41.1% versus 28.3%) and moderate and above depression (14.6% versus 10.7%). Compared to a study assessing depression among adolescents with non-HIV chronic illnesses utilizing a PHQ-9 cut off of ≥5 for a positive screen [87], we similarly found estimates in our cohort to be higher (55.7% versus 42.8%).

Our study's screening estimates are significant in that they signal the presence of a high burden of both mild and more severe CMD among APHIV who are highly engaged in HIV care. We found that double orphan status was significantly associated with increased odds of positive screens for anxiety, consistent with other studies in which bereavement has been found to be significantly associated with CMD among young people with HIV [84,88]. Although other structural and psychosocial vulnerabilities, such as school discontinuation and low resilience were not statistically significant in our study, these factors have been significantly associated with CMD in other global studies among young people with HIV [89–94]. Additionally, although viral suppression was not significantly associated with CMD in our study, studies across several settings also suggest that APHIV who struggle with viral suppression or are less engaged in HIV care may be at greater risk for experiencing moderate to severe CMD [95–97].

While several psychosocial factors were not statistically significant in multivariable models, the qualitative exploration in our study offers insights into distinct psychosocial factors related to mental health among sub-groups of APHIV, particularly those who live in CCIs and in family-based care. A substantial proportion of our participants lived in CCIs, with the rest living with a parent and/or extended families. In multivariable models, there was no significant difference in association of positive screens for CMD between CCI-based APHIV versus others. However, qualitative exploration yielded narratives that highlight divergence around mental health stressors in these sub-groups. Family support, a well-known protective factor for CMD, was often lacking in APHIV living in CCIs compared to those living with a parent and/or extended family [98–100]. Consistent with other studies [101], APHIV in CCIs received minimal family support and described strained relationships with extended family members as a source of stress. In contrast, for APHIV living with families, a parent with a shared HIV status, as well as other family members were sources of support, in addition to peers. Notably, the prospect of graduation from institutional care at age 18 years (a legal stipulation in India) and needing to navigate multiple transitions simultaneously, including transition to emerging adulthood and independent living with minimal preparation and supports, was a distinct stressor for adolescents under the age of 18 years living in CCIs.

Stigma narratives were consistent with other studies [102,103], with APHIV being highly concerned about societal and public attitudes towards people with HIV. Narratives around school discontinuation, a risk factor for CMD among youth, also centered around anticipatory and enacted stigma with examples of actual experiences of forced school exclusion

due to inadvertent disclosure of HIV status. Strikingly, in our study, public attitudes, disclosure concerns, and anticipatory and enacted stigma did not necessarily translate to negative self-image among APHIV, indicating that external stigma was much more significant than internal stigma in this population. This may in part be related to receiving peer support that incorporates regular group discussions on shared experiences of stigma and related coping mechanisms. The qualitative narratives highlight how external stigma contributes to CMD vulnerabilities among APHIV, underscoring the need for CMD interventions to be stigma-informed while also tailored to the distinct needs of sub-groups, such as orphans and those growing up in CCIs.

Onset of CMD in adolescence, in particular depression and anxiety, confers a higher risk of recurrence in adulthood [104–107]. For APHIV, addressing CMD early needs to be a priority as this has the potential to impact the trajectories of two independent, yet intersecting causes of ill-health and death–HIV and mental health disorders– across the life course in this population. A significant health system gap to address CMD exists in India and other LMICs due to sub-optimal availability of mental health professionals [108]. Task-shifted stepped-care approaches for lay-person delivery of evidence-based interventions for CMD among adolescents have been successful in school-settings in India [109–111]. However, as found in our study and others [20,112,113], youth with HIV encounter unique challenges in school settings and work settings, including school exclusion and significant concerns around disclosure. Therefore, spaces which these youth perceive as being "safe" may be better suited for task-shifted delivery of evidence-based interventions for CMD. Task-shifted approaches include leveraging existing ART counselors in public sector ART centers to screen for CMD and deliver evidence-based interventions. While high client volume at public ART centers can hinder integration of task-shifted mental health delivery roles, youth-led community-based HIV care models show promise globally [114–117] and in Indian settings [118] in addressing CMD among APHIV.

One consistent finding across studies, including ours, is the natural resonance around HIV status and lived experiences among APHIV, and consequently the preference for communication about challenges to peers. In various countries, peer supporters are part of national HIV programs, such as community treatment supporters for youth with HIV in Zimbabwe [119,120]. Recent studies indicate that peer-delivered evidence-based mental health interventions, such as problem-solving therapy, can be integrated into peer support models to address CMD [38,121,122]. Our study serves as a preliminary model that with partnership and investment, APHIV can not only provide peer support but also be trained to conduct screening for CMD among their peers. As the next step, peer-delivered interventions for CMD should be a focus of further development and research in India. Finally, developing and incorporating differentiated interventions that include additional components such as addressing externalized stigma, evidence-based ART adherence interventions [123–125], and evidence-based transition readiness tools for APHIV living in CCIs needs to be a key consideration for delivery of CMD interventions to AYPHIV [126,127].

Our study has a few limitations. First, all participants in the cross-sectional survey were recipients of the I'mPossible peer mentorship intervention, which may have mitigated CMD risk while positively influencing self-image and viral suppression. However it is less likely that peer support alone would have improved significant CMD, since no CMD-specific interventions are presently incorporated in the I'mPossible Fellowship. Second, APHIV in this study were generally highly engaged in HIV care, and we were limited in assessing CMD among those who were under-engaged in HIV care or lost to follow up, suggesting that our findings may be an underestimation of the actual prevalence among all APHIV. Moreover, since our population included only those with perinatally acquired HIV, CMD patterns may be different in youth with behaviorally acquired HIV. Additional studies utilizing population-level sampling strategies are needed to estimate true prevalence of CMD among youth with HIV across regions in India. Third, our qualitative exploration did not thoroughly explore the association of gender with CMD. In general, women in India tend to have a higher prevalence of CMD influenced by gender-specific risk factors such as gender-based violence, socioeconomic inequality, and greater caregiving burdens than men [101,102]. Fourth, the small sample size in our survey likely limited determination of statistical significance of correlates of CMD.

Our study also has several strengths. Few studies in India have utilized validated screening instruments such as the PHQ-9 or GAD-7 [128] to screen for CMD among APHIV; most studies thus far involve qualitative exploration or assess broader psychosocial outcomes such as quality of life [32,33,129–132]. Our approach of a cross-sectional survey followed by a qualitative research approach provides a more rounded understanding of CMD and their determinants among APHIV. Additionally, participatory approaches involving trained APHIV as youth investigators engaged in this research is a distinct strength, and our study is among the first in India utilizing this approach [44]. Responses to peer-delivered screening instruments are also less likely to be prone to social desirability bias, yielding more reliable estimates of positive screens for CMD.

## Conclusions

The high burden of positive screens for depression and anxiety among APHIV in our study with significant associations with orphanhood and other vulnerabilities suggest that immediate and long-term investment to integrate mental health screening and intervention implementation across the pediatric-to-adult HIV care continuum is needed for all adolescents with HIV in India. Future research should critically examine the mental health needs of those who are under-engaged in care and/or are not virally suppressed. Implementation research that incorporates community-based participatory research principles is needed to inform both screening and delivery of evidence-based interventions for CMD among APHIV. Such investments need to be a high priority as they have the potential to not only shape HIV care outcomes but also yield lasting benefits beyond HIV across the life-course for young people with HIV.

## Supporting information

**S1 Data. Consolidated sociodemographic and mental health screening dataset.**
(XLSX)

**S1 Checklist. Inclusivity in global research.**
(DOCX)

## Acknowledgments

We would like to acknowledge the contritbution of our community partners Sneha Charitable Trust and Snehagram for their support for this work.

## Author contributions

**Conceptualization:** Michael Babu Raj, Babu Seenappa, Siddha Sannigrahi, Sanya Thomas, Sunil S. Solomon, Lakshmi Ganapathi, Anita Shet.

**Data curation:** Michael Babu Raj, Babu Seenappa, Siddha Sannigrahi, Kacie Filian, Esha Nobbay, Suhas Reddy, Prashant Laxmikanth, Sanya Thomas, Lakshmi Ganapathi.

**Formal analysis:** Ashley A. Sharma, Michael Babu Raj, Siddha Sannigrahi, Kacie Filian, Suhas Reddy, Sanya Thomas, Aastha Kant, Lakshmi Ganapathi, Anita Shet.

**Funding acquisition:** Anita Shet.

**Investigation:** Ashley A. Sharma, Michael Babu Raj, Babu Seenappa, Siddha Sannigrahi, Kacie Filian, Esha Nobbay, Prashant Laxmikanth, Aastha Kant, Satish Kumar SK, Lakshmi Ganapathi, Anita Shet.

**Methodology:** Ashley A. Sharma, Michael Babu Raj, Siddha Sannigrahi, Suhas Reddy, Lakshmi Ganapathi, Anita Shet.

**Project administration:** Michael Babu Raj, Babu Seenappa, Anita Shet.

**Supervision:** Michael Babu Raj, Babu Seenappa, Satish Kumar SK, Anita Shet.

**Writing – original draft:** Ashley A. Sharma, Lakshmi Ganapathi, Anita Shet.

**Writing – review & editing:** Ashley A. Sharma, Michael Babu Raj, Siddha Sannigrahi, Satish Kumar SK, Sunil S. Solomon, Lakshmi Ganapathi, Anita Shet.

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
