## [Decision Letter · Decision Letter 0]

PMEN-D-25-00120

Mental health challenges among adolescents and young adults with perinatally acquired HIV: key findings from the I’mPossible Program in India

PLOS Mental Health

Dear Dr. Shet,

Thank you for submitting your manuscript to PLOS Mental Health and I am sorry for the delay. After careful consideration of the reviewer reports, we feel that your paper has merit but does not yet fully meet PLOS Mental Health’s publication criteria as it currently stands. Therefore, we invite you to submit a revised version of the manuscript that addresses the points raised during the review process. 

Please ensure that all reviewer comments are fully addressed - you can find them at the end of this email.

We look forward to receiving your revised manuscript.

Kind regards,

Karli Montague-Cardoso

Executive Editor

PLOS Mental Health

Journal Requirements:

Additional Editor Comments (if provided):

Reviewers' comments:

Reviewer's Responses to Questions

**Comments to the Author**

1. Does this manuscript meet PLOS Mental Health’s publication criteria?

Reviewer #1: Partly

Reviewer #2: Yes

2. Has the statistical analysis been performed appropriately and rigorously?

Reviewer #1: Yes

Reviewer #2: Yes

3. Have the authors made all data underlying the findings in their manuscript fully available (please refer to the Data Availability Statement at the start of the manuscript PDF file)?

Reviewer #1: Yes

Reviewer #2: Yes

4. Is the manuscript presented in an intelligible fashion and written in standard English?

Reviewer #1: Yes

Reviewer #2: Yes

Reviewer #1: lines 95-96: Statement could suggest that double burden exists as a function of being a LMIC, whereas I beiieve the intention is to indicate that this is where the bulk of the affected population with APHIV lives.

Lines 104-109 - missed substantive review of evidence: Bhana, A., et al. (2020). "Mental health interventions for adolescents living with HIV or affected by HIV in low- and middle-income countries: systematic review." BJPsych Open 6(5): e104.

Lines 109-110: Are there no studies in India or similar to India's context?

Lines 114-133: Description of I'm Possible Fellowship should clarify who ultimately became the participants of this study and who are the peers. How this the program interact with the child care institutions. These details appear to be in another publication, but it would be useful to provide a clearer statement regarding these aspects here.

In short the section detailing participants and study setting should be rewritten to provide clarity and approrpiate detail for a reader who is unfamiliar with previous work. Given that there appear to be several groups recruited for various purposes, it might be helpful to detail these in a flow chart that details these - otherwise its difficult to keep track of it!

Line 220: The comment that data saturation was reached for all major topics appears to belong in the analysis section.

Lines 228-231: While determining what constitues low resilience is context dependent, is there any basis for the percentile cut offs determined for this study - either in the literature or similar studies elsewhere?

Line 231-234: It is unclear why the cut off for the PHQ-9 and GAD-7 are set at such a low value. Can the author's reference validaton studies in India that indicate these to be appropriate as typically a cut off of >=10 is used for both PHQ-9 and GAD-7. I note however that in the results section, description of scores that are >=10 are also presented.

General Comment on Analysis: It may be more useful to only undertake an analysis using those scoring >=10 rather than those who scored in the mild category (often regarded as transient and self-correcting). It may also be useful to separate the analsyis of those living in CCIs relative to families. While it may reduce the sample size for the respective analysis, it could provide a clearer picture of the findings. This would potentially avoid the problem of spurious positive findings. My endorsement of the statement that the analytic methods are appropriate does not take account of the above suggestion.

Lines 323-337: Suggesting an association exists on the basis of directionality (not significant) is not warranted.

Lines 507-520 - this description is not part of results and should be in recommendations section

Lines 455: The increased odds of positive screens for anxiety only occur if mild anixety is included, but not anixety at more severe levels. The argument about directionality is possibly overstating the findings and greater caution is required to avoid over interpreting non-significant findings.

Lines497-505: GIven that no significant associations were noted for CMD and stigma, what would be the basis for this concluding statement? It is generally overstating the relationship which is potentially likely but not demonstrated in this study!

Lines 513-514: As far as I can tell, task shifting was not measured in this study.

Lines 528-530: None of the elements described in this sentence was explicitly measured. For example "proof of concept that with partnership and investment, APHIV can not only provide peer support but also be trained to conduct screening for CMD among their peers. This use of peers for CMD does not demonstrate efficacy as an approach compared to other approaches.

Lines 569: The cut off of >=10 for PHQ-9 and GAD-7 presents only a small percentage of individuals who may be depressed, anxious or both. Is this burden similar to that of depression and anxiety in the general population or in relation to youth who are orphaned but who are not HIV positive. The concluding statement should be more nuanced to reflect these complexities.

There are minor grammatical issues scattered throughout the MS. For example, Lines 144-146: Participants... Fellowship comprising 257 peers. Those aged 13 years...study.

Reviewer #2: Lines 315 and 316: The font size for the table label seems smaller than that of table 1. Please note for consistency.

Lines 431-437: The first paragraph of the ‘Discussion section’ should highlight the main findings of your research. Your current write-up speaks to the aim and objectives of your work, but the key findings not explicitly stated. This is a section of your article a reader to check to see the main findings of your work. Consider revising.

Lines 464-466 “One may anticipate that APHIV experiencing challenges with viral suppression and engagement in HIV care will have an even higher burden of moderate and above CMD severity”. Please review this statement for clarity. It sounds vague.

Line 473: Insert a comma after ‘others’

Lines 471-475: “In multivariable models, although there was no significant difference in association of positive screens for CMD between CCI-based APHIV versus others possibly owing to small sample sizes, qualitative exploration yielded narratives that highlight divergence around mental health stressors in these sub-groups”. This sentence seems long. Consider breaking it into easy-to-read sections.

Lines: 477-480: “Consistent with other studies,88 APHIV in CCIs received minimal family support and described strained relationships with extended family members as a source of stress, whereas for APHIV living with families, a parent with a shared HIV status, as well as other family members were sources of support, in addition to peers”. This sentence is long. Consider breaking it.

Lines 516-520: “While high client volume at public sector ART centers may pose challenges to integrating task-shifted roles for mental health care delivery, youth-friendly models of HIV care led by trained youth and community members show promise globally, 101–104 and in Indian settings105 in addressing CMD among APHIV”. This sentence structure is not well written. Kindly revise it. Also the sentence is long which could it difficult for reader to follow through. In general, please try and avoid long sentences in the article.

Line 524: Please change ‘are’ to ’were’. It should read in past tense.

Line 551: …higher than what?...The comparison sounds incomplete. Kindly revise.

**Do you want your identity to be public for this peer review?** For information about this choice, including consent withdrawal, please see our Privacy Policy

Reviewer #1: **Yes: ** Arvin Bhana

Reviewer #2: No

---

## [Editor Report · Decision Letter 1]

Mental health challenges among adolescents and young adults with perinatally acquired HIV: key findings from the I’mPossible Program in India

PMEN-D-25-00120R1

Dear Dr. Shet,

We are pleased to inform you that your manuscript 'Mental health challenges among adolescents and young adults with perinatally acquired HIV: key findings from the I’mPossible Program in India' has been provisionally accepted for publication in PLOS Mental Health.

Best regards,

Karli Montague-Cardoso

Staff Editor

PLOS Mental Health